# Living Coral Displays, Research Laboratories, and Biobanks as Important Reservoirs of Chemodiversity with Potential for Biodiscovery

**DOI:** 10.3390/md23020089

**Published:** 2025-02-19

**Authors:** Ricardo Calado, Miguel C. Leal, Ruben X. G. Silva, Mara Borba, António Ferro, Mariana Almeida, Diana Madeira, Helena Vieira

**Affiliations:** 1ECOMARE, CESAM, Departamento de Biologia, Universidade de Aveiro, Campus Universitário de Santiago, 3810-193 Aveiro, Portugal; miguelcleal@ua.pt (M.C.L.); rubenxavier@ua.pt (R.X.G.S.); maraborba@ua.pt (M.B.); antonioferro@ua.pt (A.F.); d.madeira@ua.pt (D.M.); 2CESAM, Departamento de Ambiente e Ordenamento, Universidade de Aveiro, Campus Universitário de Santiago, 3810-193 Aveiro, Portugal; mariana@ua.pt (M.A.); helena.vieira@ua.pt (H.V.)

**Keywords:** Alcyoniidae, bioactive compounds, biomass production, coral propagation, loss of the source, metagenomics, replicability

## Abstract

Over the last decades, bioprospecting of tropical corals has revealed numerous bioactive compounds with potential for biotechnological applications. However, this search involves sampling in natural reefs, and this is currently hampered by multiple ethical and technological constraints. Living coral displays, research laboratories, and biobanks currently offer an opportunity to continue to unravel coral chemodiversity, acting as “Noah’s Arks” that may continue to support the bioprospecting of molecules of interest. This issue is even more relevant if one considers that tropical coral reefs currently face unprecedent threats and irreversible losses that may impair the biodiscovery of molecules with potential for new products, processes, and services. Living coral displays provide controlled environments for studying corals and producing both known and new metabolites under varied conditions, and they are not prone to common bottlenecks associated with bioprospecting in natural coral reefs, such as loss of the source and replicability. Research laboratories may focus on a particular coral species or bioactive compound using corals that were cultured *ex situ*, although they may differ from wild conspecifics in metabolite production both in quantitative and qualitative terms. Biobanks collect and preserve coral specimens, tissues, cells, and/or information (e.g., genes, associated microorganisms), which offers a plethora of data to support the study of bioactive compounds’ mode of action without having to cope with issues related to access, standardization, and regulatory compliance. Bioprospecting in these settings faces several challenges and opportunities. On one hand, it is difficult to ensure the complexity of highly biodiverse ecosystems that shape the production and chemodiversity of corals. On the other hand, it is possible to maximize biomass production and fine tune the synthesis of metabolites of interest under highly controlled environments. Collaborative efforts are needed to overcome barriers and foster opportunities to fully harness the chemodiversity of tropical corals before in-depth knowledge of this pool of metabolites is irreversibly lost due to tropical coral reefs’ degradation.

## 1. Introduction

Marine ecosystems are currently facing unprecedented anthropogenic and climate-related pressures. According to a recent survey, over 1500 scientists from more than 90 nationalities and working on (natural) science, technology, engineering, and mathematics (STEM) or social science and humanities (SSH) identified overfishing, climate change (global warming, ocean acidification), pollution (plastics), and habitat loss as the main threats to the marine environment [1]. Among the most important marine ecosystems are tropical coral reefs, commonly perceived as cradles of biodiversity populated by numerous organisms of different shapes and colors. However, the value of tropical coral reefs extends beyond supporting biodiversity; they also support human populations directly by providing a source of protein and indirectly by sustaining economic activities like fishing and tourism and acting as natural barriers that protect coastlines from erosion and storm surges [2].

In the last decades, both local (e.g., overfishing, pollution, and coastal development) and global (e.g., ocean warming and acidification) stressors have led to devastating bleaching events and die-offs [3,4,5,6,7]. During marine heatwaves, the symbiotic association between the cnidarian animal host (corals) and its microalgal photoendosymbionts (zooxanthellae) is disrupted, with the latter being expelled to the water column and the coral acquiring a bleached appearance. Whenever abnormally high seawater temperatures prevail over long periods, the effects of bleaching events may be irreversible and cause the permanent loss of reef-forming photosymbiotic corals [8,9]. This loss rapidly cascades over other organisms that colonize tropical coral reefs, promoting the overall collapse of these habitats. Bleached coral reefs commonly become either barren rocky landscapes or are covered with opportunistic fast-growing macroalgae, which feed on the organic matter released by coral death retained in the reef [10]. This shift from coral to algal dominance causes significant changes to the ecosystem and results in an overall decline of biodiversity.

One of the most emblematic tropical coral reefs in the world is the Great Barrier Reef (GBR) in Australia. Covering nearly 350,000 km^2^, the GBR has been recurrently impacted by bleaching events; as marine heatwaves become more frequent and last longer than they used to, the condition of this flagship marine ecosystem is certainly of concern [11]. The effects of marine heatwaves on the GBR are further enhanced by periodic tropical cyclones and storms, along with outbreaks of bacterial and viral diseases. Unfortunately, coral reef ecosystems of the Coral Triangle (a region in the western Pacific Ocean that encompasses the waters of Indonesia, Malaysia, the Philippines, Papua New Guinea, East Timor, and the Solomon Islands), likely the most biodiverse eco-region in the globe, and those located elsewhere (e.g., the Caribbean and the Red Sea), are also experiencing the effects of climate change and major declines [3,4,7]. From 2009 to 2018, 14% of coral cover was lost across the world’s reefs [12]. The onset of a new mass bleaching event in February 2023, which continued to expand in 2024, is now the largest on record, with 77% of the world’s coral reefs estimated to suffer from bleaching-level heat stress [13]

In addition to playing a pivotal role in supporting biodiversity, all tropical regions populated by coral reef ecosystems have been the source of new marine bioactive compounds since the early 1970s [14]. A variety of terpenoids, sterols, alkaloids, and long-chain fatty acids with anti-inflammatory, antibacterial, anti-tumor, and immunosuppressive activities have been isolated, particularly from soft corals and gorgonians [15,16,17]. Indeed, the easily accessible intertidal flats and/or shallow areas of tropical coral reefs, which can be visited using snorkeling or scuba diving gear, have become the “playground” of organic chemists unraveling the chemodiversity of these marine habitats (Figure 1). Cnidarians in general, and photosymbiotic corals in particular, are well-known sources of new marine natural products [17,18]. Indeed, soft coral species within family Alcyoniidae alone were responsible for nearly one out of five new marine natural products reported in the scientific literature between 2010 and 2019 [19]. Moreover, corals harbor a plethora of microbial communities, some of which are species-specific and may change in response to stressors and bleaching events [16]. These microbial communities also produce antimicrobial compounds to eliminate pathogens from their host corals [20,21], and thus, not only corals, but also their associated microorganisms are considered extraordinary sources of bioactive natural products [16,22].

Researchers worldwide are now aware of how the loss of tropical coral reefs may translate into the permanent loss of biodiversity and chemodiversity, including, in particular, important biomolecules yet to be screened from the marine organisms inhabiting these ecosystems and that may translate into new products, processes, and services powered by blue biotechnology. Indeed, the chemodiversity of tropical coral reefs is yet to be fully tapped. It is therefore paramount to put forward actions safeguarding the valuable biomolecules present in these ecosystems so that they are not permanently lost. Public or private displays featuring live tropical corals, research laboratories working with these organisms in multiple scientific areas (e.g., ecophysiology, ecotoxicology, biochemistry, and sexual and asexual reproduction), and biobanks hosting living or preserved samples of tropical corals may ensure that, regardless of the resilience and recovery of tropical coral reefs in the wild, their chemodiversity is preserved. The present work describes how living coral displays, research laboratories, and biobanks, defined by the Organization for Economic Cooperation and Development (OECD) [23] as crucial scientific “Noah’s Arks”, may enable the continued bioprospecting of tropical coral reefs chemodiversity, even under the most catastrophic scenario of irreversible loss of these unique marine ecosystems.

## 2. Living Coral Displays

Tropical photosymbiotic corals rank among the most popular invertebrates commonly displayed as marine ornamental species, both in public and private aquaria [24]. Public aquaria first appeared in the mid-nineteenth century as displays of life occurring elsewhere or worldwide, but they have evolved to encompass education, scientific research, and conservation roles [25,26,27]. Such public displays can be found worldwide on their own in zoos and in theme parks and in dining and shopping venues with tanks holding hundreds to millions of liters water (Figure 2). As for private aquaria, several million people worldwide own a marine aquarium, and here too the goal has evolved from keeping the largest and most exotic display to establishing reef communities of corals, other invertebrates, and fishes [24]. Advances in tank design and equipment allow for setting up highly efficient life support systems that secure optimal seawater chemistry and illumination, perfectly mimicking the natural conditions experienced by tropical photosymbiotic corals in the wild, allowing them to thrive and even reproduce in both public and private displays [28]. Moreover, with premium marine salt mixes being commercially available on a regular basis (some even being specifically formulated for the husbandry of coral reef organisms) and water purification engines using reverse-osmosis being increasingly more efficient, one can easily prepare high-quality synthetic seawater anywhere, even thousands of kilometers away from the ocean.

Despite growing concerns for animal welfare and environmental conservation, aquaria featuring tropical corals still focus mainly on aesthetics, i.e., on the shapes and colors of coral reefs, as these factors are key to grasping viewers’ attention. Hence, aquaria displaying coral reef communities often stock specimens with the most conspicuous or unusual color patterns and/or species featured in movies [24] that do not necessarily inhabit the same reef area in the wild, thereby setting the stage for the production of new metabolites with biotechnological potential. For instance, different color morphs may produce contrasting metabolites, and the associations between corals and their zooxanthellae and/or microorganisms are likely to be changed due to the biochemical and cellular stresses of adapting to an aquarium environment [28,29]. Moreover, inter-specific hybridization can be performed *ex situ* using *Acropora* species, allowing for the production of hybrids that display enhanced growth rates and a greater tolerance to higher water temperatures and ocean acidification than purebred specimens [30]. Although wild coral populations have the capacity to adapt to different abiotic and biotic conditions within a certain range, their genetic diversity may be compromised in aquaria, as most coral nursery and propagation techniques are still based on the asexual propagation of coral clones (commonly through fragmentation) [31]. This is further enhanced by inbreeding and genetic drift in response to new selective pressures [32], including intra- and interspecific competition. The densely packed environment commonly experienced in aquaria by corals promotes competition for growth space, which may be claimed by chemically fighting neighboring colonies in direct contact (e.g., digestive activity) or at distance (e.g., allelopathy) [33]. While the “aggressor” coral colony will synthetize biomolecules destined to dislodge, damage, or kill the coral colony being targeted (not necessarily a different species), the “victim” coral colony will also synthetize biomolecules to mitigate the deleterious effects of this “chemical aggression” [34,35,36,37].

Altogether, the abiotic conditions of coral reef aquaria, the multitude of coral species stocked, and the symbionts associated with coral colonies may trigger the production of new metabolites that are less likely to occur in their natural habitat. If environmental conditions in the aquarium system are maintained, these new metabolites may be produced consistently [38], and their chemical diversity can be further increased by fine tuning temperature, nutrient concentrations, turbidity, and other abiotic drivers [39]. Additionally, by using the most advanced genomic and transcriptomic tools, the responsible genes/pathways for such metabolites can be identified and partially or totally isolated or reprogrammed via synthetic biology, allowing for their production in a much simpler way. Likewise, advances in aquarium technology and husbandry allow for mimicking seasonal water temperature fluctuations, solar irradiance, lunar cycles, and diel cycles experienced in the wild. Public aquaria may offer the ideal setting to produce metabolites recorded in natural coral reefs, but also allow for gathering new insights into how the genetic composition of aquarium animals and the aquarium environment introduce novel selection pressures that may lead to the production of new bioactive compounds [28]. Either way, to perform clinical trials and then commercialize therapeutic metabolites isolated and identified from corals and their associated microorganisms, kilogram-scale levels of the target compound are required. Therefore, large-scale and sustainable techniques are paramount to culture and extract such compounds from corals for pharma and nutraceuticals industries [40,41]. This requirement may be a caveat only solved using specialized systems not for displaying but rather for mass-producing coral biomass [42].

## 3. Research Laboratories

Coral gardening (taking small coral fragments and growing them until they are sexually mature) has been extensively used by scientists involved in coral reefs restoration, along with micro-fragmentation [43,44]. Micro-fragmentation relies on splitting adult corals into small clonal fragments to increase the number of individuals that can be used to restore the reef [44,45,46], either directly or after a grow-out nursery phase to achieve higher post-transplantation survival rates [47]. However, the cost-effectiveness of coral restoration using these techniques has been questioned [47,48,49], and the genetic variability of the transplanted coral populations is low, which compromises their resilience to changing climate conditions. In this sense, research efforts have been put into achieving successful sexual larval propagation *ex situ* and under controlled conditions to improve coral genetic diversity and resilience, as well as their adaptive and evolutionary potential [47]. Furthermore, techniques classified as assisted colonization, which may include moving species beyond their current boundaries, and assisted evolution, a designation that includes selective breeding, hybridization, and genetic engineering, have been developed [48]. Nevertheless, artificially selected genotypes may not present higher fitness than natural ones when living in the wild. For instance, aquarium-cultivated stony corals frequently show weaker skeletons than wild ones [50], and some species may perform worse in the aquarium than in the wild in terms of weight gain and tissue loss [36]. Moreover, the knowledge of the coral genome and how it can be engineered to produce coral reef ecosystems that resist and thrive under the expected conditions of ocean acidification and warming is still insufficient and thus not safe to use in nature [48,51,52]. However, these novel genotypes might produce new bioactive molecules *per se* or when submitted to heat-shock, UV radiation, and different pH conditions, among other environmental variations [53]. Metabolites may also differ when the coral species is isolated or living with other corals, as both intra- and interspecific competition plays a role in defining a species’ metabolome [37] under wild and culture conditions. Differences in the type of metabolites produced were found between wild and cultured soft corals (*Klyxum simplex* and *Sinularia flexibilis*) [54,55]. In addition, corals may associate with different symbionts (zooxanthellae and microorganisms) when living under different environmental conditions [41,56,57]. For instance, >99% of the symbionts in nursery-raised *Acropora palmata* genotypes were the dinoflagellate *Cladocopium goreaui*, while *Symbiodinium* “fitti” dominate wild *A. palmata* populations in the Caribbean and the Florida Keys [58]. Additionally, there are no records of *Cladocopium* sp. symbionts in wild *A. palmata* [44]. As several of the secondary metabolites isolated from corals with anticiotic activity are in fact produced by their symbionts [59,60], the new associations formed under the controlled conditions of research laboratories’ aquaria might result in new metabolites of medical/pharmaceutical interest.

Although the diversity of bioactives produced by soft corals has been the subject of several studies and reviews published since the 1970s, many publications do not detail if the coral specimens surveyed were wild types or cultivated [40]. A study published in 1974 identified new diterpenes from the Red Sea soft coral *Sarcophytum* sp. (Figure 3A) [61]; another three new diterpenes were isolated from the same species in 2013 [62]. Overall, hundreds of bioactive compounds have been isolated from corals and their symbionts. Studies on *Xenia* sp. (Figure 3B) published from 1977 to 2019 identified 199 terpenes (180 diterpenes, 14 sesquiterpenes, and 5 steroids) from both wild and aquarium-cultivated colonies [63], and in studies published until March 2019, 92 terpenoids were identified from 16 *Alcyonium* sp. corals collected worldwide [64]. The more than 100 research papers on the secondary metabolites of *Sinularia* sp. (Figure 3C) have also revealed nearly 250 new compounds, mostly terpenes, with bioactivity [59].

It remains a challenge to determine if lab-cultivated corals and their associated symbionts can consistently and sustainably source both known and novel bioactive compounds required for biotechnological applications. Although coral biomass can be enhanced by coral gardening (e.g., *Acropora cervicornis* biomass was enhanced 1.4–1.8 times after 90 days [65]) and metabolite production can be sustained consistently by maintaining stable environmental conditions in the culture system and optimizing critical parameters [40], coral biomass might not be sufficient to harvest the required metabolites. Indeed, over 90 bioactive compounds have already been isolated from coral-associated microorganisms (CAM) and not from the animal tissue of the host coral [16]. This number is expected to grow as the importance of CAM and algal symbionts in bioactive compound production is increasingly recognized; nonetheless, establishing the optimal conditions for culturing these organisms may be an even more challenging task [30]. However, techniques like reverse genomics, metagenomics, antibody-mediated isolation, genome mining, diffusion growth chambers, and microwell chip devices have been used to successfully isolate previously uncultured CAM [66,67,68], and the biomass of target microorganisms is often easier to scale up than that of marine invertebrates [69]. Hence, the bioprospection of bioactives in CAM and their subsequent mass production in CAM themselves or in easy-to-grow bacteria, via heterologous expression, opens a route for the biotechnological application of several coral-derived bioactives.

## 4. Biobanks

Biobanks were elected by *Time* magazine in 2019 as one of the ten ideas that would change the world. The reason was that they offered the opportunity to obtain essential knowledge to aid in disease treatment. Although originally defined by the Organization for Economic Cooperation and Development (OECD) as a “collection of biological material and the associated data and information stored in an organized system, for a population or a large subset of a population” [23,70], biobanks have evolved greatly from tissue and blood collections to include stem cells, genetic material, and microbiome samples, further strengthening their essential role in biomedical science. Moreover, biobanking has been defined by the International Organization for Standardization (ISO 20387:2018) as “the process of acquisitioning and storing, together with some or all of the activities related to the collection, preparation, preservation and testing, analyzing and distributing defined biological material as well as related information and data”. Interestingly, the European Commission stresses that biobanks concern the collection of biological samples and associated data organized in a systematic way for medical scientific research and diagnostic purposes only [71]. Moreover, a recent review on the principles of biobanking defined biobanks as “large collections of biospecimens linked to relevant personal and health information that are held predominantly for use in health and medical research” [72]. In this sense, collections of tissues and/or genetic materials from organisms other than humans, and/or used for other purposes, are included in biological resource centers (BRCs), biorepositories, and other collections of biospecimens (although the distinction between them is not always clear), leading to diverse interpretations and uses. In the present review, the term “biobank” is used as defined by the OECD and refers to curated biospecimen collections and their associated data *sensu lato*.

Biobanks set in academia are usually research-oriented and supported by institutional funding and grants; biobanks in industrial settings are business-oriented, aiming at obtaining a certain product, service, and/or process [73]. While the former may be obliged to provide access to their collections for research/conservation purposes, the latter may restrict access by means of high sample costs and protection of intellectual property. The cost of purchasing samples from commercial biobanks and restrictions on international shipment may further hinder open access to biobanks. Nevertheless, biobanks are a crucial source of samples and data for researchers in academia and biomedical and pharmaceutical industries, with coral biobanks being no exception. The Living Coral Biobank (https://www.foreverreef.org/ (accessed on the 2 January 2025)), for instance, was primarily established to protect the 415 species of stony corals on the Great Barrier Reef, but it also aims to expand and include all known hard corals worldwide. To feed this biobank, coral species are identified in the wild, and samples are collected and further divided into multiple smaller fragments. Each fragment is then placed on a radio frequency identification device (RFID) for database tracking and kept in a tank. In addition, tissue samples are cryopreserved, and DNA sequences are stored in a database (accessed through the RFID). This processing allows for different uses of the living organisms, preserved tissues, and genetic information for diverse research purposes, including the extraction and further synthesis of bioactive metabolites. Living coral fragments of biobanks allow for expanding the uses referred to in previous sections because of their linked information, such as precise geolocation, the coral species they interacted with in the wild, and genomic, transcriptomic, metabolomic, epigenomic, and/or proteomic data, as well as information on the symbionts they host (namely, the symbiotic microalgae and CAM living on coral mucus). Information on these relationships and on coral holobiont biology and omics may help discover or identify bioactive metabolites of interest. Indeed, the amount of omics data for coral holobionts has increased at an unprecedented pace in the last decade [74]. For instance, genome data of *Acropora millepora* allowed for identifying genes linked to stress-tolerance phenotypes [75]. Transcriptome data of the dinoflagellate symbiont *Symbiodinium microadriaticum* elucidated the role of mRNAs and microRNAs in gene regulation [76] and allowed for building a library containing 1045 full-length 16S ribosomal RNA gene sequences that were mapped to full bacterial genomes [77], as well as the identification of stress tolerance gene function.

Understanding the biological pathways with annotated genes of the host coral, symbiotic microalgae, and CAM may reveal the processes regulating the production of certain bioactive metabolites. It is certainly important to highlight that the centralization, accurate curation, and accessibility of coral biobanks are key to successfully bioprospecting new metabolites of interest. Moreover, as coral cryopreservation and *in vitro* fertilization technologies evolve to allow for generating larvae and juveniles from cryopreserved gametes for restoration purposes (e.g., the Taronga CryoDiversity Bank of the Reef Restoration and Adaptation Program) [78], research aiming to continue the exploration of bioactive compounds produced by the coral holobiont will certainly benefit from such endeavors. Cryopreserved tissues may maintain their structure and function when samples are restored to physiological temperatures and thus be available for metabolic and genomic studies. In addition, selective breeding may be applied using cryopreserved gametes with or without applying genome editing techniques, which have been successful in thermal tolerance studies [51]. Overall, these new toolboxes can pave the way to continue the bioprospecting of coral species, even if they are unable to thrive in the oceans of the Anthropocene.

## 5. Bioprospecting Strategies and Valorization

As evidenced in the previous sections, living coral displays, both public and private, corals and/or coral communities established in research laboratories, and biobanks are outstanding reservoirs of chemodiversity, hosting a plethora of metabolites that may still be unknown to science. Moreover, this chemodiversity can be harnessed for multiple biotechnological innovations, including high-end uses like pharmaceutics, nutraceuticals, and cosmeceuticals, biomaterials for both medical and tissue engineering applications, and green (bio)chemistry processes. Bioprospecting in these environments is therefore paramount to identify new compounds, improve our current understanding on already known compounds, and isolate such compounds in the amount and purity required to fuel biodiscovery pipelines. Modern bioprospecting strategies increasingly rely on less invasive and advanced technologies, such as omics and bioinformatics, which enable the identification and analysis of bioactive compounds more efficiently [79] and sustainably, as only minute amounts of biological material is required [17]. The power of synthetic biology and reverse engineering of microorganisms for bio manufacturing of target compounds can also play a decisive role in unraveling the chemodiversity of the coral holobiont [30,80]. Although bioprospecting in “artificial ecosystems”, as discussed in this review, can mitigate the ecological and ethical concerns associated with *in situ* sampling from wild coral reefs, there is still a long way to go to fulfil the potential this approach is recognized to hold. Indeed, even when housing a multitude of coral species together, living coral displays continue to be an underrepresentation of all of the biodiversity found in natural coral reef ecosystems. This lower genetic and species diversity, together with the environmental conditions set, can constrain the range and expression of bioactive compoundss available for isolation, as many are derived from complex interactions or stress responses that may be absent from, differ or emerge in the artificial setting. Moreover, many bioactive compounds associated with corals are produced by their symbiotic microorganisms, and they often require specific conditions that mimic the natural microenvironment of the coral. It may therefore not be surprising if under such highly controlled conditions, such as the ones experienced by corals on these displays, the full array of coral-associated microorganisms found in the wild may not be able to successfully thrive. Whether these potential constraints on coral microbial diversity will result in a reduced repertoire of bioactive compounds available for bioprospecting remains an unanswered question.

To successfully bioprospect for new metabolites of interest, it is of utmost importance to standardize protocols for sample collection, processing, and analysis in living coral displays, research laboratories, and biobanks. This has been a concern of researchers and innovators working in multiple life science areas, and, in response, the International Society for Biological and Environmental Repositories (ISBER) published the “ISBER Best Practices: Recommendations for Repositories” and regularly updates it [81,82,83,84,85]. Although taking place in artificial environments, bioprospecting must still comply with regulations framed by the collection, use, and sharing of genetic resources, collectively termed Access and Benefit Sharing (ABS) frameworks [86]. Navigating these regulatory landscapes can be complex, particularly when corals originate from multiple institutions or when international collaborations are involved. For instance, equitable benefit sharing (EBS) and compliance with the Nagoya Protocol on Access to Genetic Resources and the Fair and Equitable Sharing of Benefits Arising from Their Utilization to the Convention on Biological Diversity (CBD) are cornerstones of responsible bioprospecting. Equitable benefit sharing can take various forms, including monetary compensation, capacity building, technology transfer, and the sharing of research results, and models of EBS are essential for maintaining trust and cooperation between providers and users [87]. The Nagoya Protocol establishes a legal framework that governs access to genetic resources and traditional knowledge, ensuring that the rights of resource providers are respected and, like EBS, that the benefits derived from the use of genetic resources are shared fairly and equitably. As such, compliance with the Nagoya Protocol indicates that the terms of corals bioprospecting have been mutually agreed upon among stakeholders and that prior informed consent, if required, has also been obtained. For instance, research on a particular coral bioactive may produce intellectual property rights that must be considered if a patent or revenue is later generated by a private company. The Nagoya Protocol provides a clear and transparent legal framework that encourages investment and research while safeguarding the interests of all parties involved. Overall, to benefit the most from bioprospecting in living coral displays set in aquaria or research laboratories and biobanks, there is a recognized need to: (i) standardize practices for specimens and their associated data collection, storage, processing, and distribution; (ii) allow access to technological advancements; (iii) establish collaborative networks among researchers, industry stakeholders, policymakers, and local communities; and (iv) regulate and ensure compliance with ethical practices of collecting, storing, and sharing [86] (Figure 4). It is the responsibility of nations governing these genetic resources to ensure that benefits directly or indirectly generated by them are partially, or even fully, re-invested in their conservation, therefore safeguarding their use by future generations.

The translation of bioprospected compounds into commercial products and their economic valorization also present challenges. Compounds derived from corals may require synthetic production methods [30], such as cell-free synthesis and strain engineering [88], or alternative cultivation strategies [67], which can be technically challenging. For instance, disentangling the source of bioactive compounds—whether they originate from the coral, its symbionts, or associated microorganisms—often requires identifying and characterizing bioactive compounds from coral samples using omics [79], involving techniques like high-resolution mass spectrometry [89], nuclear magnetic resonance spectroscopy [90], and advanced bioinformatics [90]. These methods are resource-intensive, requiring specialized equipment and expertise, which may not be readily available in all research settings. Fostering partnerships between research institutions and private companies may help overcome this issue and aid in the production of bioactive compounds in the amount and purity required for industrial and commercial use. As the process of isolation, characterization of bioactivity, and commercialization carries financial risks and uncertainty, more awareness of the whole value chain’s processes and needs is deemed relevant to all researchers engaging in coral bioprospection. Furthermore, preliminary focused financial and economic viability studies are also required to establish the net value of producing the bioactive compounds bioprospected from corals in public and private living displays, research laboratories, and biobanks.

## 6. Concluding Remarks

Despite increasing awareness on how anthropogenic actions, particularly global carbon emissions, are negatively impacting world oceans, sadly, each year new records are set related to massive die-offs of tropical coral reefs worldwide. At present, it seems almost inevitable to permanently lose nearly 95% of all shallow-water tropical corals in the next 10 years or so. While reef-building stony corals commonly receive most media and scientific attention as the most visible face of this crisis, one cannot overlook the deleterious effects these same stressors also promote in soft corals. Collaborative efforts between researchers, industry, and policymakers are vital to overcome existing obstacles and fully unravel the potential of tropical corals as valuable reservoirs of untapped chemodiversity, thus allowing for the continuation of bioprospecting efforts to unravel the legacy of tropical corals’ secondary metabolites. Corals challenge the common concept of species, as species’ genomes do indicate that they naturally interbreed [91]. Additionally, molecular phylogenetic clades do not always agree with chemotypes or the morphological features known to be of taxonomic relevance (such as sclerites) [92]. Despite its limitations [93], DNA barcoding remains a powerful tool to reveal coral species’ richness. However, even if tropical coral species’ genetic diversity is indeed saved, certain chemotypes may be irreversibly lost if DNA barcoding alone is employed and chemodiversity is simply overlooked. Overall, conservation efforts must ensure that multidisciplinary scientific teams work closely together and safeguard the full legacy of tropical corals to ensure that it remains available for future generations.

## Figures and Tables

**Figure 1 marinedrugs-23-00089-f001:**
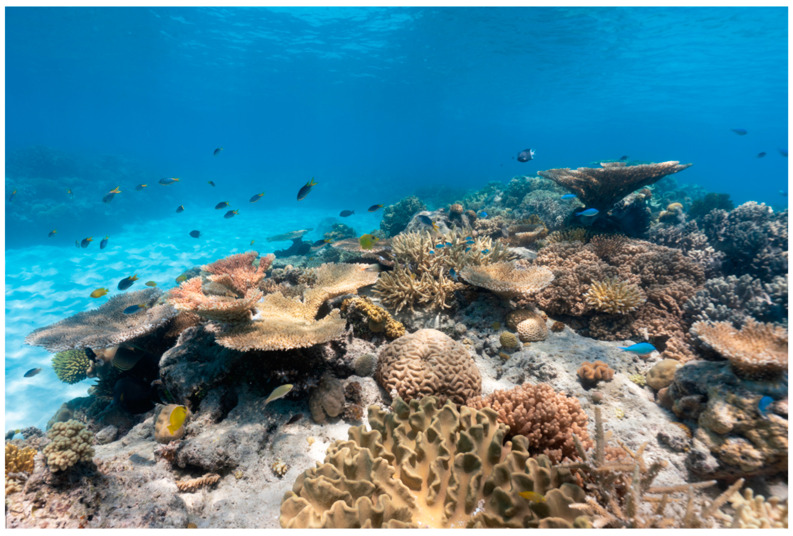
Tropical coral reef flat in the Indo-Pacific region featuring multiple colonies of stony and soft corals. Photo credits: Marcos Borsatto (Envato Elements).

**Figure 2 marinedrugs-23-00089-f002:**
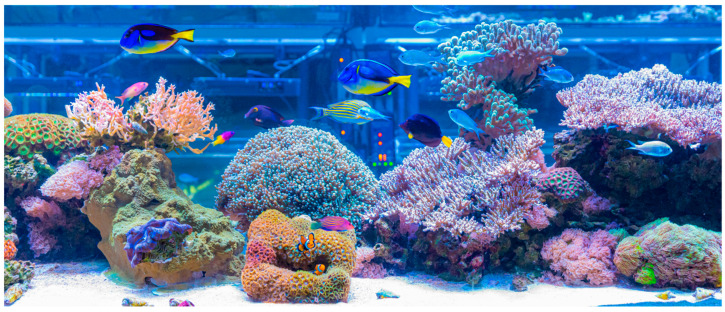
Coral reef display at CEPAM-ECOMARE (University of Aveiro, Portugal) featuring multiple colonies of stony and soft corals, along with other coral reef invertebrates and fish.

**Figure 3 marinedrugs-23-00089-f003:**
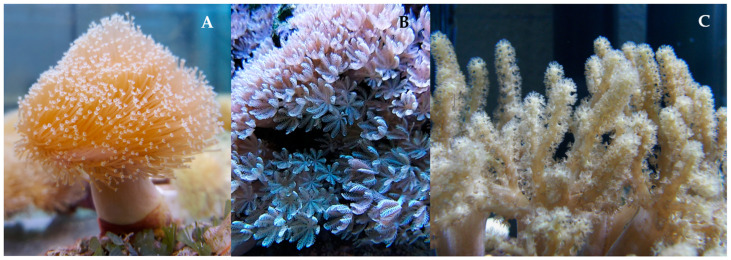
Specimens of *Sarcophytum* sp. (**A**), *Xenia* sp. (**B**), and *Sinularia* sp. (**C**) at the Blue Biobank (UA) at CEPAM-ECOMARE (University of Aveiro, Portugal).

**Figure 4 marinedrugs-23-00089-f004:**
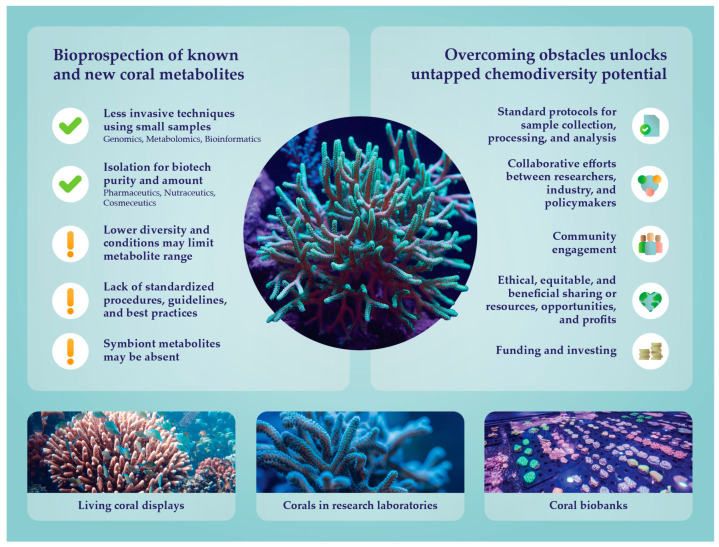
The advantages and challenges of bioprospecting living coral displays, coral communities of research laboratories, and biobanks.

## Data Availability

Not applicable.

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
