# Peer review of "Living Coral Displays, Research Laboratories, and Biobanks as Important Reservoirs of Chemodiversity with Potential for Biodiscovery"

_marinedrugs, 2025, doi:10.3390/md23020089_

Round 1

Reviewer 1 Report

Comments and Suggestions for Authors

Dear Authors,

coral reefs are highly-important ecosystems, which are not only attractive aesthetically, but also possess some biological material for further medical and other developments. The related knowledge is yet to be systematized, and, thus, I fully agree with your idea to prepare such a review. Indeed, your manuscript is suitable to the chosen journal. Indeed, it is informative, more or less well illustrated, and appropriately referenced. However, it lacks some important information and needs better organization (see my comments below). Although this work promises to be comprehensive, it is not clear whether this is really so or not. I strongly feel that your manuscript deserves publication, but only after in-depth revisions (also extensions and logical normalization).

1)      Although this is a review, the manuscript must contain two, rather extensive sections, namely Methodology, Discussion. The former should explain the principles of the information analysis for the purposes of this review (e.g., how literature was collected, how it was analyzed, etc.?). The latter should indicate research biases and practical implications. A section “Conclusions” would also help.

2)      You should always articulate where is the knowledge from the literature and where are your own ideas and interpretations. I strongly recommend to move all your own ideas and interpretations to Discussion.

3)      Fig. 1: your own photo? Give credit if not so.

4)      Your study proposes a kind of exploitation of coral reefs for finding new, potentially useful materials. If so, you should explain how this exploitation can look like physical, how should it correspond to conservation of coral reefs, and who should regulate all these activities? Again, these are good topics for Discussion.

5)      Finally, what about cold-water corals?

Generally, this work needs in-depth updates, including re-organizations and extensions. Alternatively, it can be submitted as a viewpoint (this requires consultations with the journal’s representatives), but even this solution will require some extensions and re-organizations suggested above.

Author Response

Reviewer#1 Comments and Suggestions for Authors

Responses to each reviewer comment (RxCy) (x being the number of the reviewer and y the number of the comment by the same reviewer) are provided bellow as RxRy.

The anonymous reviewers are acknowledged, as their insightful comments and constructive criticism significantly helped to improve the overall quality of the final manuscript.

R1C1: Dear Authors, coral reefs are highly-important ecosystems, which are not only attractive aesthetically, but also possess some biological material for further medical and other developments. The related knowledge is yet to be systematized, and, thus, I fully agree with your idea to prepare such a review. Indeed, your manuscript is suitable to the chosen journal. Indeed, it is informative, more or less well illustrated, and appropriately referenced.

R1R1: The authors acknowledge the positive feedback by Reviewer#1.

R1C2: However, it lacks some important information and needs better organization (see my comments below). Although this work promises to be comprehensive, it is not clear whether this is really so or not. I strongly feel that your manuscript deserves publication, but only after in-depth revisions (also extensions and logical normalization).

R1R2: We have tried to address the constructive criticism provided by Reviewer#1 in the most comprehensive way and accommodate the recommendations and corrections provided. Whenever we decided not to do so, we have put forward a solid scientific rationale clearly detailing why we have done so.

R1C3: Although this is a review, the manuscript must contain two, rather extensive sections, namely Methodology, Discussion. The former should explain the principles of the information analysis for the purposes of this review (e.g., how literature was collected, how it was analyzed, etc.?). The latter should indicate research biases and practical implications. A section “Conclusions” would also help.

R1R3: While the authors do understand the points put forward by Reviewer#1 on the more “conventional” structure of review manuscripts (featuring a Methodology and Discussion section), we respectfully disagree that these are mandatory sections on review studies. Indeed, Marine Drugs Instructions for Authors on its Manuscript Submission Overview section, namely when referring to Review articles refers that “The structure can include an Abstract, Keywords, Introduction, Relevant Sections, Discussion, Conclusions, and Future Directions.”, the words “can include” and not “must include” provide the authors the degrees of freedom necessary to put forward less conventional ways of structuring their review papers. Moreover, there are multiple Review articles published in Marine Drugs that do not feature a Methodology and Discussion section (just to refer some examples already published in 2025: https://doi.org/10.3390/md23020060, https://doi.org/10.3390/md23020061, https://doi.org/10.3390/md23020056, https://doi.org/10.3390/md23020055. A potential issue that may also help to frame this constructive criticism by Reviewer#1 is the fact that Marine Drugs only accepts either “Articles” (original research manuscripts) or “Reviews”. Marine Drugs does not feature “Opinions”, “View Points”, “Perspectives” or any other related-type of manuscripts that are more commonly perceived as personal views of the authors. It is therefore the authors opinion that this is ultimately an editorial decision that either the Editor in Chief and/or the Section Editor in Chief must take on whether the way our Review is presented is acceptable or not for publication in Marine Drugs. As such, the authors have decided to maintain the current structure of the present Review and ask the Editor in Chief and/or the Section Editor in Chief of Marine Drugs to make an editorial recommendation on this issue.

Nonetheless, we do agree with Reviewer#1 that a “Conclusions” section would help to “wrap up” our manuscript and deliver a more solid “take home message” to the reader. As such, to best accommodate this constructive criticism by Reviewer#1 we have now added section “6 Concluding Remarks” to the end of our manuscript and it reads as follows (Lines 461-481): “Despite increasing awareness on how anthropogenic actions, particularly global carbon emissions, are negatively impacting world oceans, sadly, each year new records are set of massive die-offs of tropical coral reefs worldwide. At present, it almost seems inevitable to permanently lose nearly 95% of all shallow water tropical corals in the next 10 years or so. While reef building hard corals commonly receive most media and scientific attention as the most visible face of this crisis, one cannot overlook the deleterious effects that these same stressors also promote in soft corals. Collaborative efforts between researchers, industry, and policymakers are vital to overcome existing obstacles and fully unravel the potential of corals as valuable reservoirs of untapped chemodiversity, thus allowing to continue bioprospecting efforts and unravel the legacy of tropical corals secondary metabolites. Corals challenge the common concept of species, as species genomes do indicate that they naturally interbreed [91]. Additionally, molecular phylogenetic clades do not always agree with chemotypes, neither with morphological features known to be of taxonomic relevance (such as sclerites) [92]. Despite its limitations [93], DNA barcoding remains a powerful tool to reveal coral species richness. However, even if tropical coral species genetic diversity is indeed saved, certain chemotypes may be irreversibly lost if DNA barcoding alone is employed and chemodiversity is simply overlooked. Overall, conservation efforts must ensure that multidisciplinary scientific teams work closely together and safeguard that the full legacy of tropical corals remains available for future generations.”.

R1C4: You should always articulate where is the knowledge from the literature and where are your own ideas and interpretations. I strongly recommend to move all your own ideas and interpretations to Discussion.

R1R4: The authors acknowledge the recommendation by Reviewer#1, but there is a clear way to easily tell apart what is “knowledge from the literature” and the authors “ideas and interpretations”, as the first is always supported by the scientific references cited over the manuscript and the second stand alone with no references being cited. To “pool” all the authors “ideas and interpretations” in a single section of the manuscript (a Discussion, as suggested by Reviewer#1) would disrupt our line of thought and the rationale put forward to support the structure of our manuscript. As such, the authors respectfully disagree from Reviewer#1 and will maintain the structure of our manuscript as it is, unless the Editor in Chief and/or the Section Editor in Chief of Marine Drugs instruct otherwise.

R1C5: Fig. 1: your own photo? Give credit if not so.

R1R5: We have obtained the photo with license rights from the stock bank Envato Elements. “We have revised Fig 1 legend as follows: Tropical coral reef flat in the Indo-Pacific region featuring multiple colonies of stony and soft corals. Photo credits: Marcos Borsatto (Envato Elements).”

R1C6: Your study proposes a kind of exploitation of coral reefs for finding new, potentially useful materials. If so, you should explain how this exploitation can look like physical, how should it correspond to conservation of coral reefs, and who should regulate all these activities? Again, these are good topics for Discussion.

R1R6: This certainly an important topic for manuscript as the one we put forward and we consider that these issues are extensively covered in our section “5. Bioprospecting Strategies and Valorization”, with an emphasis on the Nagoya Protocol on Access to Genetic Resources and the Fair and Equitable Sharing of Benefits Arising from Their Utilization to the Convention on Biological Diversity. This is ultimately enforced by the nations governing the rights on the genetic resources being exploited. We would like to bring Reviewer#1 attention to Lines 385 to 412, that read as follows: “Although taking place in artificial environments, bioprospecting must still comply with regulations framed by the collection, use, and sharing of genetic resources, collectively termed Access and Benefit Sharing (ABS) frameworks [86]. Navigating these regulatory landscapes can be complex, particularly when corals originate from multiple institutions or when international collaborations are involved. For instance, equitable benefit-sharing (EBS) and compliance with the Nagoya Protocol on Access to Genetic Resources and the Fair and Equitable Sharing of Benefits Arising from Their Utilization to the Convention on Biological Diversity (CBD) are cornerstones of responsible bioprospecting. Equitable benefit-sharing can take various forms, including monetary compensation, capacity building, technology transfer, and the sharing of research results, and models of EBS are essential for maintaining trust and cooperation between providers and users [87]. The Nagoya Protocol establishes a legal framework that governs access to genetic resources and traditional knowledge, ensuring that the rights of resource providers are respected and, like EBS, that the benefits derived from the use of genetic resources are shared fairly and equitably. As so, compliance with the Nagoya Protocol indicates that the terms of coral bioactives’ bioprospecting have been mutually agreed upon among stakeholders and prior informed consent, if required, has also been obtained. For instance, research on a particular coral bioactive may produce intellectual property rights that must be considered if a patent or revenue is later generated by a private company. The Nagoya Protocol provides a clear and transparent legal framework that encourages investment and research while safeguarding the interests of all parties involved. Overall, to benefit the most from bioprospecting in living coral displays set in aquaria or research laboratories and biobanks there is a recognized need to: (i) standardize practices for specimen and their associated data collection, storage, processing, and distribution; (ii) allow access to technological advancements; (iii) establish collaborative networks among researchers, industry stakeholders, policymakers, and local communities; and (iv) regulate and ensure compliance with ethical practices of collecting, storing, and sharing [86].”

To further emphasize this idea and specially refer to conservation, as well-suggested by Reviewer#1, we have added the following sentence (Lines 412-415): “It is the responsibility of nations governing these genetic resources to ensure that benefits directly or indirectly generated by them are partially, or even fully, re-invested in their conservation, therefore safeguarding their use by future generations.”

R1C7: Finally, what about cold-water corals?

R1R7: This is a very good point raised by Reviewer#1, as indeed we have failed to adequately frame our study as solely focusing on corals originating from tropical coral reefs (the word “tropical” was missing from the text, hence misleading the reader). To clarify this issue, our revised manuscript clearly frames its scope at an early stage by highlighting in the Abstract and the Introduction that it solely addresses tropical corals. Cold water corals are also facing multiple threats, and their conservation is certainly also a matter of concern. However, given the current technical advances in the maintenance and propagation of tropical corals hosting microalgal photoendosymbionts (zooxanthellae) and the generalized state of crisis that tropical coral reefs face worldwide (namely unprecedented bleaching events ravaging these ecosystems even more frequently and for longer periods), we decided to focus our work solely on tropical corals. Below please find some examples of the corrections performed:

Line 13 (Abstract): “Over the last decades, bioprospecting of tropical corals…”

Lines 33-35 (Abstract): “Collaborative efforts are needed to overcome barriers and foster opportunities to fully harness the chemodiversity of tropical corals, before the in-depth knowledge of this pool of metabolites is irreversibly lost due to tropical coral reefs degradation.”

Lines 48-51 (Introduction): “Among the most important marine ecosystems are tropical coral reefs, commonly perceived as cradles of biodiversity populated by numerous organisms of puzzling shapes and dazzling colors. However, the value of tropical coral reefs extends beyond supporting biodiversity;”

We also consider that the concluding sentence of our Introduction frames the scope of our work in a clear way, highlighting that we are solely addressing tropical corals:

Lines 118-122 (Introduction): The present work describes how living coral displays, research laboratories, and biobanks (as defined by the Organization for Economic Cooperation and Development (OECD) [23] are crucial as scientific “Noah's Arks” that may enable the continued bioprospecting of tropical coral reef chemodiversity, even under the most catastrophic scenario of irreversible loss of these unique marine ecosystems.

R1C8: Generally, this work needs in-depth updates, including re-organizations and extensions. Alternatively, it can be submitted as a viewpoint (this requires consultations with the journal’s representatives), but even this solution will require some extensions and re-organizations suggested above.

R1R8: Please refer to R1R3.

Reviewer 2 Report

Comments and Suggestions for Authors

This manuscript provides a broad but reasonably comprehensive review of the prior work and current state of understanding of coral reefs, including coral displays, research, and biobanks. These aspects of corals are presented in the context of their importance in biodiversity, chemodiversity, and the potential for the discovery of new compounds. The text is well written and logically presented, with figures and diagrams that enhance the review. There is a substantial reference section. My only criticism is that there is no detailed information on the important pharmaceuticals that have come from corals.

Author Response

Reviewer#2 Comments and Suggestions for Authors

Responses to each reviewer comment (RxCy) (x being the number of the reviewer and y the number of the comment by the same reviewer) are provided bellow as RxRy.

The anonymous reviewers are acknowledged, as their insightful comments and constructive criticism significantly helped to improve the overall quality of the final manuscript.

R2C1: This manuscript provides a broad but reasonably comprehensive review of the prior work and current state of understanding of coral reefs, including coral displays, research, and biobanks. These aspects of corals are presented in the context of their importance in biodiversity, chemodiversity, and the potential for the discovery of new compounds. The text is well written and logically presented, with figures and diagrams that enhance the review. There is a substantial reference section.

R2R1: The authors acknowledge the positive feedback by Reviewer#2.

R2C2: My only criticism is that there is no detailed information on the important pharmaceuticals that have come from corals.

R2R2: We acknowledge the constructive criticism by Reveiwer#2, but we would like to highlight that in our Introduction (lines 85 to 94) we do refer the variety of bioactive compounds isolated from corals: “A variety of terpenoids, sterols, alkaloids, and long-chain fatty acids with anti-inflammatory, antibacterial, anti-tumor, and immunosuppressive activities have been isolated particularly from soft corals and gorgonians [15–17].” and “Cnidarians in general, and photosymbiotic corals in particular, are well-known sources of new marine natural products [17,18]. Indeed, soft coral species within family Alcyoniidae alone were responsible for nearly one out of five new marine natural products reported on the scientific literature between 2010 – 2019 [19].”.

We further address this topic on section “3. Research Laboratories” (lines 233 to 240): “Overall, hundreds of bioactive compounds have been isolated from corals and their symbionts. Studies on Xenia sp. (Figure 3B) published from 1977 to 2019 identified 199 terpenes (180 diterpenes, 14 sesquiterpenes, and five steroids) from both wild and aquarium-cultivated colonies [63], and in studies published until March 2019, 92 terpenoids were identified from 16 Alcyonium sp. corals collected worldwide [64]. The more than 100 research papers on the secondary metabolites of Sinularia sp. (Figure 3C) have also revealed nearly 250 new compounds, mostly terpenes, with bioactivity [59].”

Two of the authors of the present work (Calado and Leal) have recently published a review on Marine Drugs (please see [19] 19. Calado, R. et al. Updated Trends on the Biodiscovery of New Marine Natural Products from Invertebrates. Mar. Drugs 2022, 20, 389) reporting how corals have yielded such a great diversity of new bioactive compounds over the last decades. We think that providing and extensive list of such compounds would go beyond the scope of the present study and end-up deviating the reader from our main take home message. Nevertheless, if the Editor in Chief and/or the Section Editor in Chief of Marine Drugs consider that adding this information is paramount for this work, the authors will do their best to accommodate this suggestion.

Reviewer 3 Report

Comments and Suggestions for Authors

This review promotes the use of public aquariums, research laboratories, and biobanks of corals with potential for the production of marine drugs. This is a noble idea that may work in an ideal world, but there are some practical implications that need more attention. One issue is that the ms addresses examples referring to scleractinian corals that may not be relevant for soft corals. The other one is that soft corals may be difficult to identify, while different species produce different secondary metabolites and therefore the role of species identification needs more attention.

Specific comments

Line 20. “Coral livings displays provide …  ”. Is this correct grammar?

Line 47. “puzzling shapes and dazzling colors”. Please, use less dramatic wording.

Line 57. “gaining a bleached appearance”. Incorrect choice of words. Gaining is a benefit, not a loss.

Lines 94, 244, 310. “coral host” should be “host coral”, similar to “host species”.  

Line 84. The introduction of the paper is discussing threats to corals in general, but most of these concern reef-building scleractinians, while corals with the highest bioprospecting potential are soft corals. Although many soft corals occur on coral reefs, these different roles (stony corals as dominant reef builders and soft corals as most promising providers of bioactive compounds) should be more clear. So, the proportion of how many species of stony corals and how many species of soft corals are known to produce secondary metabolites that are relevant for the production of marine drugs? This distinction really needs to be addressed more clearly because otherwise readers may get a wrong impression.

Lines 177-183. Bioprospecting usually requires large quantities of organism. Since public aquariums are not equipped for this, what kind of facilities will be provided to solve this problem? 

Lines 192-194. Reproduction by fragmentation results in poor genetic diversity and cultivation by sexual propagation is not yet possible at an industrial scale. Since many coral species are involved, how will this problem be solved?

Line 212-216. This example would only be relevant if the scleractinian A. palmata is known to produce secondary metabolites. Please add information on such secondary metabolites or explain better why A. palmata is used as an example here because it is not representative for soft corals such as Klyxum simplex and Sinularia flexibilis.

Line 221. This sentence evokes the question what would be the diversity of bioactives produced by stony corals.

Line 239. This example ccncerns a scleractinian with a heavy skeleton. What can be said about soft corals?

Lines 304-309. Again, this is an example of a stony coral, but how is this example relevant for soft corals?

Line 332. The relationship with bioconstruction is not clear. Please provide a reference. Perhaps one can be found in the topical collection “Biogenic reefs at risk” of the journal “Frontiers in Marine Science”

Author Response

Reviewer#3 Comments and Suggestions for Authors

Responses to each reviewer comment (RxCy) (x being the number of the reviewer and y the number of the comment by the same reviewer) are provided bellow as RxRy.

The anonymous reviewers are acknowledged, as their insightful comments and constructive criticism significantly helped to improve the overall quality of the final manuscript.

R3C1: This review promotes the use of public aquariums, research laboratories, and biobanks of corals with potential for the production of marine drugs. This is a noble idea that may work in an ideal world, but there are some practical implications that need more attention.

R3R1: The authors acknowledge the positive feedback by Reviewer#3.

R3C2: One issue is that the ms addresses examples referring to scleractinian corals that may not be relevant for soft corals.

R3R2: Indeed we agree with Reviewer#3 that some specific issues addressed will most likely affect scleractinian corals. Nonetheless, several anthropogenic pressures, such as those that will ultimately lead to bleaching events, will also negatively affect soft corals and other marine invertebrates hosting microalgal photoendosymbionts (e.g., sea anemones, jellyfishes, giant clams…). If Reviewer#3 specifically refers parts of our manuscript where this issue should be emphasized, the authors will certainly try to best accommodate such recommendations.

R3C3: The other one is that soft corals may be difficult to identify, while different species produce different secondary metabolites and therefore the role of species identification needs more attention.

R3R3: We fully agree with Reviewer#3. Not only is species identification challenging for soft corals, it is also challenging for scleractinian corals (please see Colin, L., et al. (2022). What's left in the tank? Identification of non-ascribed aquarium's coral collections with DNA barcodes as part of an integrated diagnostic approach. Conservation Genetics Resources 14(2): 167-182.). It is known that specimens from the same species may be remarkably different from a chemical point of view (one coral species may feature multiple chemotypes, challenging the “classical” concept of species). To stress this issue we have added the following information in section “6 Concluding Remarks” (Lines 472-478): “Corals challenge the common concept of species, as species genomes do indicate that they naturally interbreed [91]. Additionally, molecular phylogenetic clades do not always agree with chemotypes, neither with morphological features known to be of taxonomic relevance (such as sclerites) [92]. Despite its limitations [93], DNA barcoding remains a powerful tool to reveal coral species richness. However, even if tropical coral species genetic diversity is indeed saved, certain chemotypes may be irreversibly lost if DNA barcoding alone is employed and chemodiversity is simply overlooked.”

R3C4: Line 20. “Coral livings displays provide …  ”. Is this correct grammar?

R3R4: Thank you for this remark. The sentence should read “Living coral displays provide…”. It has now been corrected accordingly.

R3C5: Line 47. “puzzling shapes and dazzling colors”. Please, use less dramatic wording.

R3R5: We have reworded this part of the text (Line 50) and it now reads “of different shapes and colors.”

R3C6: Line 57. “gaining a bleached appearance”. Incorrect choice of words. Gaining is a benefit, not a loss.

R3R6: As suggested, we have corrected the wording and it now reads “the coral acquiring a bleached appearance.”.

R3C7: Lines 94, 244, 310. “coral host” should be “host coral”, similar to “host species”. 

R3R7: Corrected as suggested throughout the manuscript.

R3C8: Line 84. The introduction of the paper is discussing threats to corals in general, but most of these concern reef-building scleractinians, while corals with the highest bioprospecting potential are soft corals. Although many soft corals occur on coral reefs, these different roles (stony corals as dominant reef builders and soft corals as most promising providers of bioactive compounds) should be more clear. So, the proportion of how many species of stony corals and how many species of soft corals are known to produce secondary metabolites that are relevant for the production of marine drugs? This distinction really needs to be addressed more clearly because otherwise readers may get a wrong impression.

R3R8: Reviewer#3 raises an interesting question that we consider to be transversal to future readers of our work and most likely results from the way available literature on the bioprospecting of corals inhabiting tropical coral reefs presents findings on this topic. While soft corals have indeed been the main sources of new marine natural products (please see [18] and [19] authored by some of the co-authors of the present manuscript), this does not mean that hard corals feature a lower chemodiversity. The surveys supporting these publications rely on databases that report the species from which a given compound is reported for the first time ever in scientific literature; this does not mean that such compound does not occur in any other species. Researchers preferably screen soft corals over scleractinian corals because these are much easier to process in the laboratory and, despite their water content, the amount of living tissue yielded from soft corals is commonly higher (namely because there is no need to strip the thin layer of living tissue from the carbonate skeleton, which can be a time-consuming task). We may add a few words on this issue in our Introduction if Reviewer#3 (or the Editor in Chief and/or Section Editor) requires so, but we do not think it is paramount; in the present work we address both soft and scleractinian corals as a whole, as we consider that all tropical corals face an unprecedented state of crisis, although soft corals remain largely “under the radar”, receiving much less media and scientific attention than they should.

R3C9: Lines 177-183. Bioprospecting usually requires large quantities of organism. Since public aquariums are not equipped for this, what kind of facilities will be provided to solve this problem?

R3R9: We acknowledge Reviewer#3 point of view, but we respectfully disagree. At present, bioprospecting pipelines targeting marine invertebrates require much lower amounts of biomass than they used to at the early years of biodiscovery. The biomass required to supply such pipelines can easily be produced in public aquariums (or research centres) through asexual propagation using fragmentation (please see R3R10 below). Most public aquariums run coral propagation programs on a regular basis, so they can share among them different species and/or morphotypes without having to remove new coral colonies from the wild. Moreover, they commonly have to “trim” their coral reef displays to avoid some species to overgrow others, commonly generating a surplus of biomass from certain coral species on display that can be used for screening and further propagated to generate more biomass if necessary. These coral colonies may also be shared with research centres, commonly for scientific goals only, so their production can be upscaled if necessary. It is also worth referring that more and more public aquariums are now venturing into the sexual propagation of corals (see R3R10 below), making available even higher yields of biomass from some coral species whose clonal production through fragmentation is less interesting to pursue conservation goals.

R3C10: Lines 192-194. Reproduction by fragmentation results in poor genetic diversity and cultivation by sexual propagation is not yet possible at an industrial scale. Since many coral species are involved, how will this problem be solved?

R3R10: Indeed, as well remarked by Reviewer#3, asexual propagation through coral fragmentation results in poor genetic diversity. In fact, it produces genetic clones of a given mother colony. However, this can be an advantage when aiming to produce biomass of a given coral genotype, as asexual propagation safeguards that exact genetic copies of the target specimen will be replicated. The advantages of this approach for bioprospecting natural compounds from corals have already been advocated; please see reference [38] (Leal, M.C. et al. Coral Aquaculture to Support Drug Discovery. Trends Biotechnol. 2013, 31, 555–561). Moreover, it may even allow to replicate the microbiome of mother colonies through in toto culture (see Pimentel, T., et al. (2016) Bacterial communities from corals cultured ex situ remain stable under different light regimes — Relevance for in toto aquaculture. Aquaculture 450: 258-261; and Leal, M. C., et al. (2014). Marine Microorganism-Invertebrate Assemblages: Perspectives to Solve the "Supply Problem" in the Initial Steps of Drug Discovery. Marine Drugs 12(7): 3929-3952.). Additionally, new breakthroughs in the sexual propagation of corals have also paved the way for the large-scale production of these organisms (namely scleractinian corals), using standardized production systems and protocols (please see Craggs, J., et al. (2017). Inducing broadcast coral spawning ex situ: Closed system mesocosm design and husbandry protocol. Ecology and Evolution 7(24): 11066-11078) and Randall, C. J., et al. (2020). Sexual production of corals for reef restoration in the Anthropocene. Marine Ecology Progress Series 635: 203-232.). As such, we consider that our current knowledge on both sexual and asexual reproduction of corals can indeed allow to produce the biomass required to fuel the initial stages of the biodiscovery pipelines for new natural products derived from these organisms (or their associated microbiome).

R3C11: Line 212-216. This example would only be relevant if the scleractinian A. palmata is known to produce secondary metabolites. Please add information on such secondary metabolites or explain better why A. palmata is used as an example here because it is not representative for soft corals such as Klyxum simplex and Sinularia flexibilis.

R3R11: All coral species studied to date are known to produce secondary metabolites, regardless of being soft or scleractinian corals. Soft corals are more commonly targeted to identify new bioactive compounds, but this does not mean that other coral species do not produce them (please see R3R8). Such findings are simply not reported in scientific literature because of their “lack of novelty”. We selected this specific example using a widely propagated coral species (A. palmata) due to its endangered status in the wild, as it perfectly illustrates how contrasting the microbiome associated with wild and cultured specimens of the same coral species can be. If Reviewer#3 could please detail how we can improve this part of our manuscript for clarity, the authors will certainly address any corrections required.

R3C12: Line 221. This sentence evokes the question what would be the diversity of bioactives produced by stony corals.

R3R12: Again, we honestly apologize to Reviewer#3, as the authors cannot understand the issue being raised with this comment. If Reviewer#3 could please elaborate on how we can address the concern expressed, the authors will certainly perform the corrections required to best accommodate this recommendation.

R3C13: Line 239. This example concerns a scleractinian with a heavy skeleton. What can be said about soft corals?

R3R13: If we well-understood the issue raised by Reviewer#3, it concerns the viability of large amounts of soft coral biomass being produced in captivity to support drug discovery. If so, yes, the same rationale is also valid for soft corals with protocols being available now for a few years (please see Khalesi M. et al. (2008) The soft coral Sinularia

flexibilis: potential for drug development. In: Leewis R, Janse M (eds) Advances in Coral Husbandry in Public Aquariums, pp. 47–60. Burgers’ Zoo, Arnehm, the Netherlands.).

R3C14: Lines 304-309. Again, this is an example of a stony coral, but how is this example relevant for soft corals?

R3R14: Again, if we well-understood the issue raised by Reviewer#3, it concerns the viability of using an approach such as The Living Coral Biobank that essentially targets scleractinian corals and uses RFID tags to identify each specimen in the biobank. Technically, there is no limitation on why such approach (tagging with RFID) would not be possible with soft corals, as the tag can be easily secured using a rubber band, nylon string or any other material, with the soft corals ending up to overgrow the material attaching the RFID tag and even, most likely overgrowing the tag (which would ultimately be embedded in the coral tissue, but still allow the screening of that colony using an RFID reader).

R3C15: Line 332. The relationship with bioconstruction is not clear. Please provide a reference. Perhaps one can be found in the topical collection “Biogenic reefs at risk” of the journal “Frontiers in Marine Science”

R3R15: The authors acknowledge that this terminology may not be familiar to all readers. For clarity, we have eliminated the term “bioconstruction” from the sentence as it does not disrupt its content. It now reads as follows (Lines 351-354): “Moreover, this chemodiversity can be harnessed for multiple biotechnological innova-tions, including high-end uses like pharmaceutics, nutraceuticals and cosmeceuticals, biomaterials for both medical and tissue engineering applications, as well as green (bio)chemistry processes.”.

Round 2

Reviewer 1 Report

Comments and Suggestions for Authors

Dear Authors,

thanks for your revisions and responses! Although I'd prefer review papers looking a bit differently, I agree that it is the journal's editor who should make such judgments. So, I agree that your contribution is reasonable in its present form.

Reviewer 3 Report

Comments and Suggestions for Authors

Line 35:  "tropical coral reef degradation" (without s)

I am disappointed that the authors have not tried more to address my concerns about the confusion between soft corals and stony corals. Stony corals are of ecological importance but not for the extraction of bioactive compounds. Now it seems that the authors just like to promote their project and publish a paper about it for without considering whether it will be useful to the readers.